# ADV3D: Generating Safety-Critical 3D Objects through Closed-Loop Simulation

**Jay Sarva**[1,3†] **Jingkang Wang**[1,2] **James Tu**[1,2] **Yuwen Xiong**[1,2]
**Sivabalan Manivasagam**[1,2] **Raquel Urtasun**[1,2]

Waabi[1]    University of Toronto[2]    Brown University[3]
jay_sarva@brown.edu {jwang,jtu,yxiong,smanivasagam,urtasun}@waabi.ai

**Abstract:** Self-driving vehicles (SDVs) must be rigorously tested on a wide range of scenarios to ensure safe deployment. The industry typically relies on closed-loop simulation to evaluate how the SDV interacts on a corpus of synthetic and real scenarios and verify it performs properly. However, they primarily only test the system's motion planning module, and only consider behavior variations. It is key to evaluate the full autonomy system in closed-loop, and to understand how variations in sensor data based on scene appearance, such as the shape of actors, affect system performance. In this paper, we propose a framework, ADV3D, that takes real world scenarios and performs closed-loop sensor simulation to evaluate autonomy performance, and finds vehicle shapes that make the scenario more challenging, resulting in autonomy failures and uncomfortable SDV maneuvers. Unlike prior works that add contrived adversarial shapes to vehicle roof-tops or roadside to harm perception only, we optimize a low-dimensional shape representation to modify the vehicle shape itself in a realistic manner to degrade autonomy performance (e.g., perception, prediction, and motion planning). Moreover, we find that the shape variations found with ADV3D optimized in closed-loop are much more effective than those in open-loop, demonstrating the importance of finding scene appearance variations that affect autonomy in the interactive setting. Please refer to our project page https://waabi.ai/adv3d/ for more results.

**Keywords:** Closed-loop simulation, Adversarial robustness, Self-driving

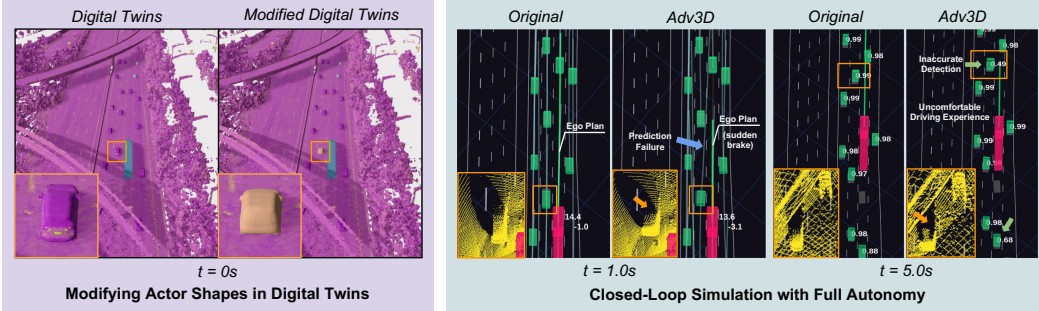

Figure 1: ADV3D is a framework that evaluates autonomy systems in closed-loop on complex real-world scenarios, and alters the scene appearance via actor shape modification (left) to create more challenging scenarios that cause autonomy failures such as inaccurate detections, incorrect predictions and strong decelerations or swerving, resulting in uncomfortable and dangerous maneuvers (right).

## 1   Introduction

Most modern autonomy systems in self-driving vehicles (SDVs) perceive the agents in the scene, forecast their future motion, and then plan a safe maneuver [1, 2, 3, 4]. These tasks are typically

---

†Work done while a research intern at Waabi.

7th Conference on Robot Learning (CoRL 2023), Atlanta, USA.

referred to as perception, prediction and motion planning respectively. To deploy SDVs safely, we must rigorously test the autonomy system on a wide range of scenarios that cover the space of situations we might see on the real world, and ensure the system can respond appropriately. Two strategies are employed to increase scenario coverage; either synthetic scenarios are created or situations are retrieved from a large collection of logs captured during real-world driving.

The self-driving industry then relies on closed-loop simulation of these scenarios to test the SDV in a reactive manner, so that the effects of its decisions are evaluated in a longer horizon. This is important, as small errors in planning can cause the SDV to brake hard or swerve. However, coverage testing is typically restricted to the behavioral aspects of the system and only motion planning is evaluated. This falls short, as it does not consider scene appearance and ignores how perception and prediction mistakes might result in safety critical errors with catastrophic consequences, e.g., false positive detections or predictions might result in a hard break, false negatives could cause collision.

In this paper, we are interested in building a framework for testing the full autonomy system. We focus on a LiDAR-based stack as it is the most commonly employed sensor in self-driving. Naively sampling all the possible variations is computationally intractable, as we need to not only cover all the possible behavioral scenarios but also scene appearance variations, such as actor shape, the primary factor for LiDAR point clouds. Towards this goal, we propose a novel adversarial attack framework, ADV3D, that searches over the worst possible actor shapes for each scenario, and attacks the full autonomy system, including perception, prediction and motion planning (Fig. 1). This contrasts with existing approaches that focus on building a universal perturbation to a template mesh that is added into all scenes as a roof-top or road-side object, and only attacks the perception system [5, 6, 7, 8].

We leverage a high-fidelity LiDAR simulation system that builds modifiable digital twins of real-world driving snippets, enabling closed-loop sensor simulation of complex and realistic traffic scenarios at scale. Thus, when modifying the scene with a new actor shape, we can see how the autonomy system would respond to the new sensor data as if it were in the real world (e.g., braking hard, steering abruptly) as the scenario evolves. To ensure the generated actor shapes are realistic, during optimization we constrain the shape to lie within a low-dimensional latent space learned over a set of actual object shapes. This contrasts with existing approaches that search over the vertices of the mesh directly [5, 6, 7, 8], resulting in unrealistic shapes that need to be 3D-printed to exist.

In our experiments, we find ADV3D can generate challenging actor shapes on over 100 real-world highway driving scenarios and on multiple modern autonomy systems and attack configurations, resulting in perception and prediction failures and uncomfortable driving maneuvers. Importantly, we show that by attacking in closed-loop we can discover worse situations (i.e, more powerful attacks) than attacking in open loop. We believe this finding is very significant and will spark a change in the community to find scenario variations that are challenging to every aspect of autonomy.

## 2 Related Work

**Self-Driving Systems:** One of the earliest approaches to self-driving autonomy was an end-to-end policy learning network [9] that directly outputs control actuations from sensor data. This autonomy approach has significantly evolved due to advancements in network architectures, sensor inputs, and learning methods [9, 10, 1, 11, 12], but lacks interpretability. Most modern self-driving autonomy systems in industry typically break down the problem into three sequential tasks: instance-based object detection [13, 14, 15, 16], motion forecasting [17, 18, 19, 20], and planning [21, 22, 23, 24]. Some works conduct joint perception and prediction (P&P) first [25, 26]. Other works leverage shared feature representations for simultaneous perception, prediction and planning learning [27, 28]. Lately, interpretable neural motion planners have emerged, enabling end-to-end learning while ensuring modularity and interpretability [27, 2, 4]. Recently, another autonomy paradigm is instance-free autonomy that estimate spatial-temporal occupancies [2, 29, 30, 4]. To demonstrate generalizability, this work evaluates an instance-based [26] and instance-free [31] autonomy system.

**Adversarial Robustness for Self-Driving:** Research on adversarial robustness has obtained substantial attention [32, 33, 34, 35, 36, 37, 38, 39, 40, 41, 42, 43], particularly in safety-critical domains such as self-driving. Recent works primarily consider individual subsystems in self-driving. Specifically, researchers have focused on generating undetectable or adversarial objects [5, 7, 8, 44], spoofing LiDAR points [45, 46] and creating adversarial vehicle textures within simulation environments [47, 48]. Most recently, some works focus on prediction robustness, exploring data-

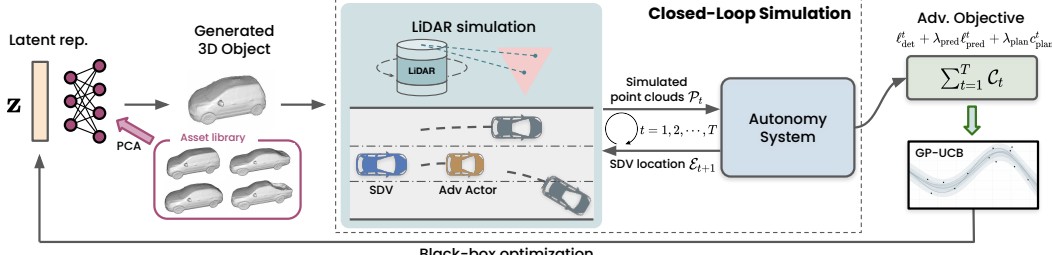

Figure 2: **Overview of ADV3D for safety-critical 3D object generation in closed-loop.** Given a real-world scenario, Adv3D modifies the shapes of selected actors, and runs LiDAR simulation and full autonomy stack in closed-loop to optimize the shape representation with black-box optimization.

driven trajectory prediction models and proposing a series of adversarial attack and defense mechanisms [49, 50, 51]. Other works evaluate planners given the ground-truth perception information [52, 53, 54, 55] or consider simplified image-based imitation learning systems [56, 57]. Finally, some works consider the entire end-to-end autonomy stack [58, 59] by modifying the actor trajectories to induce poor downstream planning performance. However, all these attacks are performed in the *open-loop setting* that are not suitable for real-world autonomy testing.

**Closed-Loop Adversarial Attacks:** The gap between open-loop and closed-loop testing for deep neural networks has been studied in [60], showing the results of open-loop testing does not necessarily transfer to closed-loop environments. Existing closed-loop adversarial attacks [61, 56, 62, 63] are performed in game-engine environments like CARLA [64]. Specifically, [61] perform adversarial attacks on an image-based autonomy by painting black lines on the road. [56, 62] attack the entire autonomy stack by modifying the trajectories of traffic participants. [63] take simple parameterized scenarios and create adversarial scenarios by optimizing the parameters. However, these works only consider a few hand-crafted synthetic scenarios with limited realism and diversity, which generalizes poorly to the real world [65]. In contrast, ADV3D generates adversarial actor shapes with a realistic closed-loop sensor simulator on over 100 real-world scenarios for a LiDAR-based autonomy.

## 3 Generating Safety-Critical 3D Objects via Closed-Loop Simulation

We aim to extend closed-loop autonomy system testing to not only include behaviors, but also scene appearance, such as actor shape variations. Towards this goal, we conduct black-box adversarial attacks against the full autonomy stack to modify actor shapes in real-world scenarios to cause system failures. To efficiently find realistic actor shapes that harm performance, we parameterize our object shape as a low-dimensional latent code and constrain the search space to lie within the bounds of actual vehicle shapes. Given a real-world driving scenario, we select nearby actors and modify their actor shapes with our generated shapes. We then perform closed-loop sensor simulation and see how the SDV interacts with the modified scenario over time and measure performance. Our adversarial objective consists of perception, prediction, and planning losses to find errors in every part of the autonomy. We then conduct black-box optimization per scenario to enable testing of any autonomy system at scale. Fig. 2 shows an overview of our approach.

In what follows, we define the closed-loop simulation setting and our attack formulation (Sec 3.1). We then describe how we build realistic scenes from real world data, parameterize the adversarial actor geometries, and carry out realistic LiDAR simulation for autonomy testing. (Sec 3.2). Finally, we present the adversarial objective and the black-box optimization approach for ADV3D (Sec 3.3).

### 3.1 Problem Formulation

**Closed-loop Autonomy Evaluation:** We now review closed-loop evaluation of an autonomy system. A traffic scenario $\mathcal{S}_t$ at snapshot time $t$ is composed of a static background $\mathcal{B}$ and $N$ actors: $\mathcal{S}_t = \{\{\mathcal{A}_t^1, \mathcal{A}_t^2, \cdots, \mathcal{A}_t^N\}, \mathcal{B}\}$. Each actor $\mathcal{A}_t$ consists of $\{\mathcal{G}, \xi_t\}$, where $\mathcal{G}$ and $\xi_t$ represent the actor's geometry and its pose at time $t$. Given $\mathcal{S}_t$ and the SDV location $\mathcal{E}_t$, the simulator $\psi$ generates sensor data according to the SDV's sensor configuration. The autonomy system $\mathcal{F}$ then consumes the sensor data, generates intermediate outputs $\mathcal{O}$ such as detections and the planned

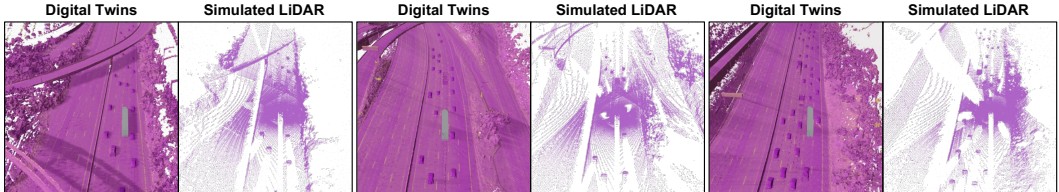

Figure 3: **Reconstructed digital twins and simulated LiDAR for real-world highway scenarios**.

trajectory, and executes a driver command $\mathcal{D}$ (e.g., steering, acceleration) via the following mapping $\mathcal{F} : \psi(\mathcal{S}, \mathcal{E}) \rightarrow \mathcal{D} \times \mathcal{O}$, The simulator then will update the SDV location given the driver command $\mathcal{E}_t, \mathcal{D} \rightarrow \mathcal{E}_{t+1}$ as well as update the actor locations to generate the scenario snapshot $\mathcal{S}_{t+1}$. This loop continues and we can observe the autonomy interacting with the scenario over time.

**Adversarial Shape Attacks in Closed Loop:**   Given our closed-loop formulation, we can now select actors in the scene to modify their shapes and optimize to find autonomy failures. For simplicity, we describe our formulation with a single modified actor, but our approach is general and we also demonstrate modifying multiple actors in Sec 4. Let $\mathcal{G}^i_{\text{adv}}$ denote the safety-critical 3D object geometry for the interactive actor $\mathcal{A}^i$. In this paper, our goal is to generate $\mathcal{G}^i_{\text{adv}}$ that is challenging to the autonomy system $\mathcal{F}$. We define a cost function $\mathcal{C}_t$ that takes the current scene and autonomy outputs to determine the autonomy performance at time $t$. We accumulate the cost over time to measure the overall closed-loop performance, resulting in the following objective:

$$\mathcal{G}^i_{\text{adv}} = \arg \max_{\mathcal{G}^i} \sum_{t=1}^{T} \mathcal{C}_t \left( \mathcal{S}_t, \mathcal{F} \left( \widetilde{\psi}(\mathcal{S}_t, \mathcal{G}^i, \mathcal{E}_t) \right) \right), \tag{1}$$

where the sensor simulator $\widetilde{\psi}$ takes the original scenario $\mathcal{S}_t$ but replaces the geometry of actor $\mathcal{A}^i$ with the optimized shape $\mathcal{G}^i$, and simulates new sensor data.

## 3.2   Realistic Sensor Simulation and Adversarial Shapes

We now describe how we build digital twins from real world scenarios to perform realistic closed-loop sensor simulation (via $\widetilde{\psi}$) for autonomy testing. We then explain how we parameterize our actor shape $\mathcal{G}^i$ to enable realistic and efficient optimization.

**Building Digital Twins for LiDAR Simulation:**   Data-driven sensor simulation [66, 67, 68, 69, 70, 71] has witnessed great success in recent years. Following [72, 73], we leverage real-world LiDAR data and object annotations to build aggregated surfel meshes (textured by per-point intensity value) for the virtual world. We manually curated a set of actor meshes that have complete and clean geometry, together with CAD assets purchased from TurboSquid [74] to create a rich asset library to model actor shapes for benign actors in the scene. During simulation, we query the actor geometries from the curated set based on the actor class and bounding box. We can then place the actor geometries according to their pose $\xi_t$ in the scenario snapshot at time $t$ based on $\mathcal{S}_t$, and then perform primary raycasting [72, 73, 75, 76] based on the sensor configuration and SDV location to generate simulated LiDAR point clouds for the autonomy system via $\widetilde{\psi}$. Fig. 3 shows examples of reconstructed digital twins and simulated LiDAR point clouds.

**Adversarial Shape Representation:**   With our digital twin representation that reconstructs the 3D world, we can now model the scenario snapshot $\mathcal{S}_t$. We now select an actor to modify its shape. To ensure the actor shape affects autonomy performance, we select the closest actor in front of the SDV as $\mathcal{A}^i$ and modify its shape $\mathcal{G}^i$ to be adversarial. To ensure the actor shape is realistic and is not a contrived and non-smooth mesh, we parameterize the geometry $\mathcal{G}$ using a low-dimensional representation $\mathbf{z}$ with realistic constraints. Specifically, for vehicle actors, the primary actor in driving scenes, we take inspiration from [77] and learn a shape representation over a large set of CAD vehicle models that represent a wide range of vehicles (e.g., city cars, sedans, vans, pick-up trucks). For each CAD vehicle shape, we first compute a volumetric truncated signed distance field (SDF) to obtain a dense representation. We then apply principal component analysis [78] (PCA) over the flattened volumetric SDF $\Phi \in \mathbb{R}^{|\mathbf{L}| \times 1}$ of the whole CAD dataset to obtain the latent representation:

$$\mathbf{z} = \mathbf{W}^\top (\Phi - \boldsymbol{\mu}), \quad \mathcal{G}(\mathbf{z}) = \text{MarchingCubes} \left( \mathbf{W} \cdot \mathbf{z} + \boldsymbol{\mu} \right) \tag{2}$$

where $\boldsymbol{\mu} \in \mathbb{R}^{|\mathbf{L}| \times 1}$ is the mean volumetric SDF for the meshes and $\mathbf{W}$ is the top $K$ principle components. We then extract the explicit mesh using marching cubes [79]. Note that each dimension in $\mathbf{z}$ controls different properties of the actors (*e.g.*, scale, width, height) in a controllable and realistic manner [77, 80]. To ensure during optimization the latent code $\mathbf{z}$ lies within the set of learned actor shapes and is realistic, we normalize the latent code to lie within the bounds of the minimum and maximum values of each latent dimension over the set of CAD models.

### 3.3 Adversarial 3D Shape Optimization

With the simulation system $\widetilde{\psi}$ and a black-box autonomy system $\mathcal{F}$ under test, we can perform closed-loop testing (Sec.3.1). Given the selected actor $\mathcal{A}^i$ and low-dimensional shape representation $\mathbf{z}$, the next step is to optimize $\mathbf{z}$ such that the overall autonomy performance drops significantly. We now introduce the adversarial objective and the search algorithms to produce safety-critical shapes.

**Adversarial Objective:** We consider an adversarial objective that accounts for the entire autonomy stack, aiming to concurrently reduce the performance of each module. Taking instance-based modular autonomy as an example, the adversarial objective $\mathcal{C}_t$ combines the perception loss $\ell_{\text{det}}$, prediction loss $\ell_{\text{pred}}$ and planning comfort cost $c_{\text{plan}}$ to measure the overall autonomy performance. We increase perception errors by decreasing the confidence / Intersection of Union (IoU) for the true positive (TP) detections and increasing the confidence / IoU for false positive (FP) proposals. For motion forecasting, we compute the averaged displacement error (ADE) for multi-modal trajectory predictions. As the modified actor shape can also affect the perception and motion forecasting of other actors, we consider the average objective for all actors within the region of interest across all the frames. In terms of planning, we calculate two costs that measures the comfort (jerk and lateral acceleration) of the driving plan at each snapshot time $t$. Formally, we have

$$\mathcal{C}_t = \ell_{\text{det}}^t + \lambda_{\text{pred}}\ell_{\text{pred}}^t + \lambda_{\text{plan}}c_{\text{plan}}^t, \tag{3}$$

$$\ell_{\text{det}}^t = -\alpha \sum_{\text{TP}} \text{IoU}(B^t, \hat{B}^t) \cdot \text{Conf}(\hat{B}^t) + \beta \sum_{\text{FP}} (1 - \text{IoU}(B^t, \hat{B}^t)) \cdot \text{Conf}(\hat{B}^t), \tag{4}$$

$$\ell_{\text{pred}}^t = \frac{1}{K} \sum_{i=1}^{K} \sum_{h=1}^{H} \left\| g_j^{t,h} - p_j^{t,h} \right\|_2, \quad c_{\text{plan}}^t = c_{\text{jerk}}^t + c_{\text{lat}}^t, \tag{5}$$

where $B^t, B^t$ are the ground truth and detected bounding boxes at time $t$ and $\alpha, \beta$ are the coefficients to balance TP and FP objectives. $g_j^{t,h}, p_j^{t,h}$ are the $h$-th ground truth and predicted waypoints for actor $j$ at time $t$ and $H$ is the prediction horizon. Lastly, $c_{\text{jerk}}^t$ and $c_{\text{lat}}^t$ represent the jerk $(m/s^3)$ and lateral acceleration $(m/s^2)$ cost at time $t$. We aggregate these costs over time $\sum_{t=1}^{T} \mathcal{C}_t$ to get the final closed-loop evaluation cost. Note that our approach is general and can find challenging actor shapes for any autonomy system. We detail in supp. the objective function for an instance-free autonomy.

**Black-box Search Algorithm:** We apply black-box optimization since we aim to keep ADV3D generic to different modern autonomy systems (including non-differentiable modular autonomy systems). Inspired by existing works [56, 58], we adopt Bayesian Optimization [81, 82] (BO) as the search algorithm with Upper Confidence Bound [83] (UCB) as the acquisition function. Since the adversarial landscape is not locally smooth, we use standard Gaussian process with a Matérn kernel. We also compare BO with the other popular search algorithms including grid search [84] (GS), random search [85, 86, 87] (RS) and blend search (BS) [88]. We also compare to a baseline that conducts brute-force search (BF) over the curated asset library.

## 4 Experiments

We showcase applying ADV3D to generate safety-critical 3D actor shapes for autonomy system testing on real-world scenarios. We first introduce the experimental setting in Sec 4.1. Then in Sec 4.2, we demonstrate the importance of adversarial optimization through closed-loop simulation for the whole autonomy stack. We further show the realism of our adversarial shape representation. Finally, we investigate different attack configurations in closed-loop including attacks with multiple actors or reactive actors, and benchmark various black-box optimization algorithms.

| Closed-Loop Test | Perception and Prediction | | | | Planning | | Execution | |
|---|---|---|---|---|---|---|---|---|
| | AP / Recall (%) ↑ | | ADE ↓ | | Planning Comfort ↓ | | Driving Comfort ↓ | |
| | AP | Recall | minADE | meanADE | Lat. $(m/s^2)$ | Jerk $(m/s^3)$ | Lat. $(m/s^2)$ | Jerk $(m/s^3)$ |
| *Autonomy-A: Instance-based [26] + [24]* | | | | | | | | |
| Original | 88.2 | 89.4 | 2.14 | 4.90 | 0.203 | 0.336 | 0.194 | 0.331 |
| Adv. open-loop | 88.3 | 89.8 | 2.08 | 4.87 | 0.214 | 0.378 | 0.207 | 0.337 |
| Adv. closed-loop | **80.1** | **84.8** | **2.40** | **5.09** | **0.263** | **0.427** | **0.265** | **0.401** |

| Closed-Loop Test | Occupancy (%) ↑ | | Flow Grounded ↑ | | Planning Comfort ↓ | | Driving Comfort ↓ | |
|---|---|---|---|---|---|---|---|---|
| | mAP | Soft-IoU | mAP | Soft-IoU | Lat. $(m/s^2)$ | Jerk $(m/s^3)$ | Lat. $(m/s^2)$ | Jerk $(m/s^3)$ |
| *Autonomy-B: Instance-free [31] + [24]* | | | | | | | | |
| Original | 83.1 | 50.4 | 94.6 | 61.2 | 0.256 | 0.319 | 0.263 | 0.315 |
| Adv. open-loop | 85.7 | 53.2 | 96.3 | 65.5 | 0.260 | 0.451 | 0.279 | 0.424 |
| Adv. closed-loop | **78.8** | **45.9** | **90.1** | **55.7** | **0.302** | **0.456** | **0.308** | **0.431** |

Table 1: **Evaluation of adversarial objects in the *closed-loop* setting.**

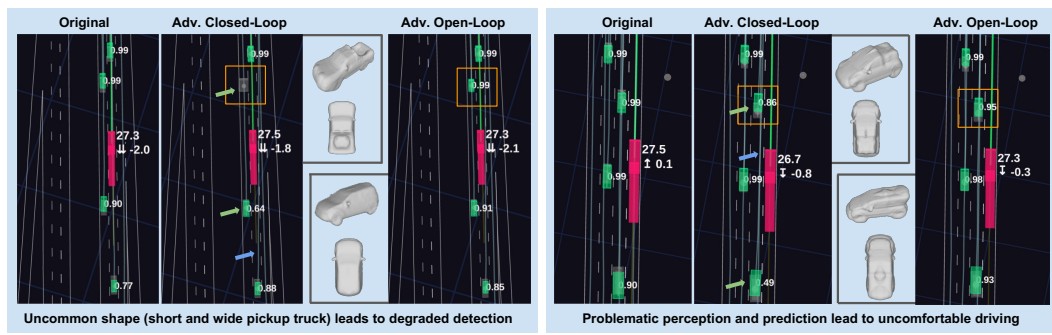

Figure 4: **Qualitative examples of adversarial shape generation in closed loop vs open loop.** We highlight the modified actors using orange bboxes and show generated adversarial shapes aside. The perception and prediction failures are pointed out by green and blue arrows. Specifically, the green arrows point to the green detected bounding boxes, and blue arrows point to the predicted light blue trajectories. The numbers on top of green predicted bounding boxes are the detection confidence scores. The numbers beside the pink ego-truck denote the current velocity and acceleration.

## 4.1 Experimental Setup

**Dataset:** We evaluate our method on a real-world self-driving dataset *HighwayScenarios*, which contains 100 curated driving snippets captured on a US highway each with a duration of 20 seconds (200 frames, sampled at 10hz). The dataset is collected with multiple LiDARs that we then annotate spatial-temporal actor tracks and leverage to build the digital twin backgrounds and reconstructed assets. It covers a wide variety of map layouts, traffic densities, and behaviors.

**Autonomy Systems:** We evaluate two state-of-the-art interpretable multi-LiDAR based autonomy systems. The first **(Autonomy-A)** uses an instance-based joint perception and prediction (P&P) model [26] which outputs actor bounding boxes and trajectories in birds-eye-view (BEV). The second **(Autonomy-B)** uses an instance-free (occupancy-based) P&P model [31] that predicts the implicit BEV occupancy and flow for queried spatial points at current (detection) and future (prediction) timestamps. Both systems use a sampling-based motion planner [24] which samples lane-relative trajectories and chooses the best trajectory with the lowest combined safety and comfort costs. Specifically, We report evaluation results for two autonomy systems in Table 1. For the rest experiments, we only study the instance-based Autonomy-A.

**Metrics:** To investigate the effectiveness of ADV3D on downstream tasks, we measure various metrics to evaluate the module-level performance, including detection, prediction and planning. For Autonomy-A, we use Average Precision (AP) and Recall to measure the detection performance. For prediction, we measure Average Displacement Error (ADE) to quantify the performance of trajectory forecasting. For Autonomy-B, we follow [31] to report mAP and Soft-IoU to measure the performance of occupancy and flow prediction. All metrics are averaged across the simulation horizon ($T = 5s$) within ROI. We report the planning comfort metrics including lateral acceleration (Lat.) and Jerk, which assess the smoothness of the proposed ego plan. To evaluate how the SDV executes in the closed loop, we evaluate system-level metrics driving comfort (*i.e.,* lateral acceleration and jerk on the executed 5s SDV trajectory). Finally, we report Jensen–Shannon divergence [89] (JSD) to measure the realism of the generated actor shapes.

| #ID | Perception $\sum_t \ell^t_{\text{det}}$ | Prediction $\sum_t \ell^t_{\text{pred}}$ | Planning $\sum_t c^t_{\text{plan}}$ | AP↑ (%, @0.5) | Recall↑ (%, @0.5) | minADE↓ $L_2$ error | meanADE↓ $L_2$ error | Lat.↓ $(m/s^2)$ | Jerk↓ $(m/s^3)$ |
|---|---|---|---|---|---|---|---|---|---|
| Original | | | | 88.7 | 89.4 | 2.51 | 4.99 | 0.261 ● | 0.294 |
| $\mathcal{M}_1$ | ✓ | | | 69.6 ● | 71.4 ● | 1.97 | 5.02 | 0.239 | 0.310 |
| $\mathcal{M}_2$ | | ✓ | | 83.1 ● | 89.1 | 2.92 ● | 6.34 ● | 0.254 | 0.412 ● |
| $\mathcal{M}_3$ | | | ✓ | 86.7 | 88.3 ● | 2.94 ● | 6.03 ● | 0.324 ● | 0.434 ● |
| **Ours** | ✓ | ✓ | ✓ | 75.4 ● | 76.4 ● | 2.82 ● | 6.21 ● | 0.411 ● | 0.410 ● |

Table 2: **Adversarial optimization for the full autonomy system.** We compare ADV3D with three baselines that each only attack one module: detection ($\mathcal{M}_1$), prediction ($\mathcal{M}_2$) and planning ($\mathcal{M}_3$). Each baseline adopts the same pipeline as ADV3D except the adversarial objective is changed. ADV3D generates actor shapes that are challenging to all downstream modules. Interestingly, it results in more uncomfortable driving maneuvers (*i.e.*, worst lateral acceleration). We mark the methods with best performances using gold ●, silver ●, and bronze ● medals.

| Algorithms | AP↑ (%, @0.5) | Recall↑ (%, @0.5) | minADE↓ $L_2$ error | Jerk↓ $(m/s^3)$ | JSD |
|---|---|---|---|---|---|
| Original | 98.7 | 99.6 | 4.70 | 0.090 | – |
| VD: $0.05m$ | 98.7 | 99.6 | 5.07 | 0.090 | 0.061 |
| VD: $0.1m$ | 98.7 | 99.6 | 5.73 | 0.090 | 0.125 |
| VD: $0.1m$ | 98.7 | 99.6 | 5.73 | 0.090 | 0.125 |
| VD: $0.2m$ | 98.7 | 99.6 | 5.78 | 0.090 | 0.285 |
| VD: $0.5m$ | 80.2 | 83.0 | 6.00 | 0.090 | 0.688 |
| VD: $1.0m$ | 46.6 | 49.8 | 7.20 | 0.163 | 0.796 |
| Adv3D (ours) | 50.3 | 55.8 | 7.87 | 0.111 | 0.175 |

Table 3: Compare with vertex deformation (VD).

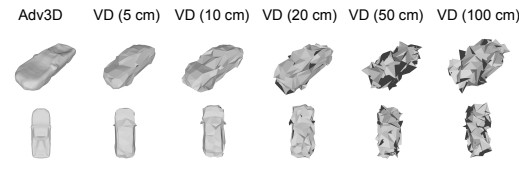

Figure 5: Qualitative comparisons with VD.

## 4.2 Experimental Results

**ADV3D finds challenging actor shapes:** We report autonomy performance metrics on the 100 traffic scenarios with and without our adversarial shape attack in Table 1. For each scenario, we set the attack search query budget to 100 queries. Our approach can degrade the subsystem performance and execution performance significantly. We further provide qualitative examples of optimized adversarial shapes together with autonomy outputs in Fig. 4. In Fig. 4 left, ADV3D successfully creates a uncommon actor shape (short and wide truck) that degrades the detection performance (low confidence or mis-detection). Fig. 4 right shows another example where the closed-loop attack finds a tiny city-car causes inaccurate detection and prediction for an actor behind the SDV and results in the SDV applying a strong deceleration. Note that the modified actor shape alters the simulated LiDAR such that perception and prediction outputs are harmed even for other actors in the scene.

**Importance of Closed-Loop Simulation:** We also compare our approach, where we find challenging actor shapes during closed-loop autonomy evaluation, against optimizing actor shapes in the open-loop setting. In the open-loop shape attack, the ego vehicle follows the original trajectory in the recorded log, and we optimize the actor shape with the same objective in Eq. 3. We then test the optimized actor shapes in closed-loop simulation where the ego vehicle is controlled by the autonomy model. Generating adversarial objects in closed-loop simulation yields substantially worse performance compared to open-loop. This indicates that it is insufficient to study adversarial robustness in open-loop as the attacks do not generalize well when the SDV is reactive. In Fig. 4, the optimized shapes in the open-loop attack do not harm autonomy performance significantly.

**Attacking Full Autonomy Stack:** Our adversarial objective takes the full autonomy stack into account. To demonstrate the importance of attacking the full system, we choose 10 logs from HighwayScenario and compare with three baselines inspired by existing works that each only attack one module (detection: [5, 7], prediction: [49, 50], and planning: [59, 90]) for Autonomy-A. To reduce performance variability, for each scenario, we perform 3 attacks, each with a budget of 100 queries, and modify the closest 3 actors in front of the SDV individually and report the worst-case performance. As shown in Table 2, attacking each downstream module produces challenging objects that are only risky to that module. In contrast, our model effectively balances all tasks to generate worst-case 3D objects that challenge the entire autonomy stack, serving as a holistic tool for identifying potential system failures. We report additional objective combinations in supp.

**Latent Asset Representation:** We also demonstrate that optimizing with our latent asset representation is more realistic than prior-works that perform per-vertex deformation. We select a scenario where the modified actor mesh from ADV3D has a strong attack against the autonomy. We initialize

| | Perception ↑ (%) | | | | Prediction ↓ | | Planning ↓ | |
| | IOU / Confidence | | AP / Recall (@0.5) | | ADE | | Planning Comfort | |
| | IoU | Conf. | AP | Recall | minADE | meanADE | Lat. $(m/s^2)$ | Jerk $(m/s^3)$ |
|---|---|---|---|---|---|---|---|---|
| Original | 69.2 | 88.8 | 92.7 | 88.7 | 2.51 | 4.99 | 0.261 | 0.294 |
| $m = 1$ | 62.9 | 71.1 | 76.5 | 79.3 | 2.83 | 6.33 | 0.312 | 0.405 |
| $m = 2$ | 61.7 | **70.1** | 71.8 | 75.1 | 2.96 | 6.30 | 0.352 | 0.460 |
| $m = 3$ | 55.7 | 77.0 | 70.2 | 73.7 | 3.37 | 6.27 | 0.360 | 0.475 |
| $m = 5$ | **50.1** | 70.9 | **64.5** | **67.7** | **3.39** | **6.34** | **0.377** | **0.535** |

Table 4: Adversarial optimization with multiple actors. $m$ denotes the number of modified actors.

| Algorithms | AP ↑ (%, @0.5) | minADE ↓ $L_2$ error | Jerk ↓ $(m/s^3)$ | #Query | GPU Hour |
|---|---|---|---|---|---|
| Original | 98.7 | 4.70 | 0.090 | – | 0.2 |
| GS [84] | 52.7 | 5.98 | 0.090 | 243 | 47.8 |
| RS [85, 86, 87] | 52.4 | 6.10 | 0.090 | 500 | 98.3 |
| BS [88] | 52.4 | 6.09 | 0.090 | 100 | 19.6 |
| BO [82] | **50.3** | 7.87 | **0.111** | 100 | 19.6 |
| Brute-Force | 52.7 | **7.91** | 0.090 | 746 | 146.6 |

Table 5: Different black-box search algorithms.

| Settings | AP ↑ (%, @0.5) | minADE ↓ $L_2$ error | Jerk ↓ $(m/s^3)$ |
|---|---|---|---|
| Original | 98.7 | 5.91 | 0.168 |
| Adv. open-loop | 59.3 | 5.23 | 0.140 |
| Adv. closed-loop | **52.1** | 6.86 | **0.335** |

Table 6: Attacks with reactive actors.

the mesh by decoding the latent **z** to generate a mean shape with 500 faces and 252 vertices, and add perturbations to vertices. The perturbations to the vertex coordinates are constrained by an $\ell_\infty$ norm, and we experiment with 4 variations $[0.1m, 0.2m, 0.5m, 1.0m]$. Results in Tab. 3 show that very loose constraints of $1m$ are needed to achieve the strong attack performance. However, the generated assets with loose constraints are unrealistic compared to ADV3D-generated meshes. We hypothesize this is because it is difficult to optimize such high-dimensional vertex-based shape representations, as black-box optimization methods suffer from curse of dimensionality.

**Multiple Adversarial Actors:**    In Table 4, we study how ADV3D can easily scale to modify the shapes of multiple actors in the scenario. We use the same 10 logs as Table 2. We set $m = [1, 2, 3, 5]$ and sample the closest actors in front of the ego vehicle. Modifying multiple actors' shapes simultaneously yields stronger adversarial attacks.

**Attack Configurations:**    We benchmark other black-box search algorithms and brute-force search in Table 5. For brute-force baseline, we iterate over all shapes in the asset library and select the one that results in the strongest attack. Table 5 shows that Bayesian Optimization (BO) leads to the strongest adversarial attack while also using the least compute. We further investigate different attack configurations on one selected log in Table 6. To increase the realism of our setting, we adopt *reactive actors* by using a traffic model [91] to control their behaviors. As shown in Table 6, ADV3D also generates challenging actor shapes in this setting and the results demonstrate the importance of closed-loop simulation.

## 5   Limitations and Conclusion

ADV3D's main limitation is that we do not optimize the actor behaviors like prior work [58, 59, 56] to allow for more diverse adversarial scenario generation. Moreover, how to incorporate ADV3D-generated safety-critical objects to create new scenarios for robust training remains future study. While our shapes are more realistic than prior work, we also observe occasionally convergence to shapes that have artifacts or oblong wheels. Better shape representations (including for non-vehicle classes) and optimization approaches (e.g., multi-objective optimization), can help create higher-fidelity and more diverse adversarial objects more efficiently.

In this paper, we present a closed-loop adversarial framework to generate challenging 3D shapes for the full autonomy stack. Given a real-world traffic scenario, our approach modifies the geometries of nearby interactive actors, then run realistic LiDAR simulation and modern autonomy models in closed loop. Extensive experiments on two modern autonomy systems highlight the importance of performing adversarial attacks through closed-loop simulation. We hope this work can provide useful insights for future adversarial robustness study in the closed-loop setting.

## Acknowledgement

We sincerely thank the anonymous reviewers for their insightful suggestions. We would like to thank Yun Chen, Andrei Bârsan and Sergio Casas for the feedback on the early results and proofreading. We also thank the Waabi team for their valuable assistance and support.

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

# Appendix

We provide additional details on our method, implementation and experimental setups, and then show additional quantitative / qualitative results. We first detail how we implement ADV3D including how to build digital twins for realistic LiDAR simulation (Sec A.1), how to create a low-dimensional representation space (Sec A.2) for adversarial shapes, the details of two modern autonomy models (Sec A.3), the adversarial optimization procedure (Sec A.4) and more experimental details. Finally then provide additional results and analysis in Sec B including full closed-loop and open-loop results for major tables in the main paper and additional experiments and additional qualitative examples. Additionally, we include a supplementary video in `https://waabi.ai/adv3d/`, providing an overview of our methodology, as well as video results on generated adversarial shapes and how it affect the autonomy performance in different scenarios.

# A    ADV3D Implementation Details

## A.1    Realistic LiDAR Simulation

Following [72, 73], we leverage real-world LiDAR data and object annotations to build surfel meshes (textured by per-point intensity value) for the virtual world. For complete background coverage, we drove through the same scene to collect multiple sets of driving data and then unify multiple LiDAR sweeps to a standard map coordinate system. We then aggregate LiDAR points from all frames and apply a dynamic point removal algorithm [92] to keep only static points and reconstruct the background $\mathcal{B}$. For dynamic actors, we aggregate the LiDAR points within object-centric coordinate bounding boxes for each labeled driving snippet. We then symmetrize the aggregated points along the vehicle's heading axis for a more complete shape. Given the aggregated points, we then estimate per-point normals from 200 nearest neighbors with a radius of 20cm and orient the normals upwards for flat ground reconstruction, outwards for more complete dynamic actors. We downsample the LiDAR points into $4cm^3$ voxels and create per-point triangle faces (radius $5cm$) according to the estimated normals. Due to sparse observations for most aggregated surfel meshes, we manually curated a set of actor meshes that have complete and clean geometry, together with CAD assets purchased from TurboSquid [74] for a larger asset variety.

## A.2    Adversarial Shape Representation

To ensure realism and watertight manifolds, we use all the CAD cars (sedan, sports car, SUV, Van, and pickup trucks) to build the low-dimensional representation. We first rescale all actors to be in a unit cube with a pre-computed scaling factor ($1.1\times$ largest dimension for all actors). Then we convert the input meshes to volumetric signed distance fields (SDF) with a resolution of $100$ (*i.e.*, $|\mathbf{L}| = 100^3$) using a open-source library[2]. Then we apply principal component analysis [78] on the flattened SDF values $\Phi \in \mathbb{R}^{|\mathbf{L}| \times 1}$ to obtain the latent representation. Specifically, we use $K = 3$ principle components for constructing the latent space. In practice, we find the larger number we use $K$, the high-frequency details can be captured but the interpolated shapes can be less realistic. This is because the non-major components usually capture the individual details instead of shared properties across all vehicles.

During optimization, we first obtain the minimum and maximum latent range $\mathbf{z}_{\min} \in \mathbb{R}^3$, $\mathbf{z}_{\max} \in \mathbb{R}^3$, where the minimum and maximum value for each latent dimension is recorded. Then we optimize a unit vector $\bar{z} \in \mathbb{R}^5$, where the first three dimensions are for PCA reconstruction, and the last two dimension indicates the scale value for width and length (range from 0.8 to 1.3). Then we normalize this first three dimension to $[\mathbf{z}_{\min}, \mathbf{z}_{\max}]$. Given the optimized latent $\mathbf{z}$, we apply Equation (2) in the main paper to get the generated 3D SDF volumes and then extract the meshes using marching cubes [79] algorithm. The extracted meshes are then scaled to the real-world size and placed in the virtual world for simulation.

---

[2] `https://github.com/wang-ps/mesh2sdf`

## A.3 LiDAR-Based Autonomy Details

Both autonomy systems tested consist of two parts, where the first part uses different joint perception and prediction models, and the second part share the same rule-based planner [24]. We assume perfect control where the ego vehicle can follow the planned trajectory (kinematic bicycle model) exactly. In other words, the driver command policy brings the ego vehicle to the next state (10Hz) by executing current planning trajectory for 0.1s. Taking instance-based autonomy for example, given the LiDAR points, the perception and prediction models output the actor locations at current and future timestamps. Then the sampling-based motion planner [24] produces the optimal plan that follows the kinematic bicycle model [93]. All the models are trained on highway driving datasets.

**Autonomy-A (Instance-Based):** We implement a variant of joint P&P model [26] to perform instance-based joint detection and trajectory prediction. For the 3D object detection part, we use a modified two-stage PIXOR [13] model following [94] which takes voxelized LiDAR point clouds as input and outputs the BEV bounding box parameters for each object. For the trajectory prediction part, we use a model that takes lane graph and detection results as input and outputs the per-timestep endpoint prediction for each object. The prediction time horizon is set to 6 seconds.

**Autonomy-B (Instance-Free):** We also verify our method on an instance-free autonomy system [31] for joint detection and motion forecasting to show its generalizability. Specifically, we replace the P&P model used in Autonomy-A with the occupancy-based model, which performs non-parametric binary occupancy prediction as perception results and flow prediction as motion forecasting results for each query point on a query point set. The occupancy and flow prediction can serve as the input for the sampling-based planner to perform motion planning afterwards.

## A.4 Adversarial Optimization Details

**Adversarial Objectives:** The adversarial objective is given in Eqn. (3-5) in the main paper, where we set $\lambda_{\mathrm{pred}} = 0.1$ and $\lambda_{\mathrm{plan}} = 0.5$ for Autonomy-A. The adversarial objective for Autonomy-B also includes three terms: $\ell_{\mathrm{det}}$, $\ell_{\mathrm{pred}}$ and $\ell_{\mathrm{plan}}$, and we keep $\ell_{\mathrm{plan}}$ as is since the PLT model we used is the same as Autonomy-A. However, the ImplicitO model in Autonomy-B does not have instance-level bounding box results, and the confidence score as well as IoU terms are no longer applicable. We thus follow [31, 95] and use the Soft-IoU metric to assess occupancy predictions. Similarly, we use the foreground mean end-point error (EPE) to measure the average L2 flow error at each occupied query point as done in [31], as the instance-based trajectory prediction is not available. Formally, the adversarial objective for Autonomy-B is defined as:

$$\mathcal{C}_t = \ell_{\mathrm{det}}^t + \lambda_{\mathrm{pred}}\ell_{\mathrm{pred}}^t + \lambda_{\mathrm{plan}}c_{\mathrm{plan}}^t, \tag{6}$$

$$\ell_{\mathrm{det}}^t = -\frac{\sum_{\mathbf{q}\in\mathcal{Q}} o(\mathbf{q})\hat{o}(\mathbf{q})}{\sum_{\mathbf{q}\in\mathcal{Q}}(o(\mathbf{q}) + \hat{o}(\mathbf{q}) - o(\mathbf{q})\hat{o}(\mathbf{q}))}, \tag{7}$$

$$\ell_{\mathrm{pred}}^t = \frac{1}{\sum_{\mathbf{q}\in\mathcal{Q}} o(\mathbf{q})}\sum_{\mathbf{q}\in\mathcal{Q}} o(\mathbf{q})||\mathbf{f}(\mathbf{q}) - \hat{\mathbf{f}}(\mathbf{q})||_2, \quad c_{\mathrm{plan}}^t = c_{\mathrm{jerk}}^t + c_{\mathrm{lat}}^t, \tag{8}$$

where $\mathcal{Q}$ is the query point set, $o(\mathbf{q})$ and $\hat{o}(\mathbf{q})$ are ground truth and predicted binary occupancy value $\in [0, 1]$ on the query point $\mathbf{q}$, respectively. Flow vector $\mathbf{f} : \mathbb{R}^3 \to \mathbb{R}^2$ and the corresponding prediction $\hat{\mathbf{f}}$ specifies the BEV motion of any agent that occupies that location. We set $\lambda_{\mathrm{pred}} = 1.0$ and $\lambda_{\mathrm{plan}} = 0.5$ for Autonomy-B.

**Black-box Optimization Details:** To handle different modern autonomy systems and the non-differentiable LiDAR simulator, we adopt the black-box optimization in ADV3D. Inspired by existing works [56, 58, 61], we adopt the Bayesian Optimization [81, 82] (BO) as the search algorithm, which maintains a surrogate model and select the next query candidate based on historical observations and acquisition function. Specifically, we use a standard Gaussian process (GP) model with Upper Confidence Bound [96, 83] (UCB) as the acquisition function. We set the exploration multiplier $\beta = 1.0$ to balance exploitation and exploration. Since the adversarial landscape is not locally smooth, we use the Matérn 3/2 kernel (product over each dimension with a length scale of 0.1) for the GP model. Unless stated otherwise, we set the total query budget as 100 and the first 11 queries are used for the initialization.

We also benchmark the other popular black-box algorithms including grid search [84] (GS), random search [85, 86, 87] (RS) and blend search (BS) [88]. For GS, we set 3 search points per dimension thus in total $3^5 = 243$ queries. For a random search, we set the query budget as 500 to achieve better performance. BS is an economical hyperparameter optimization algorithm that combines local search and global search. We adopt the official implementation[3]. We also compare a baseline that conducts brute-forcing (BF) over the curated asset library with 746 vehicles and find the worst-case actor shape. Our optimization pipeline is built on the Ray Tune framework [97].

## A.5 Additional Experimental Details

**Realism Evaluation for Generated Shapes:** We evaluate the realism of ADV3D using Jensen–Shannon divergence [89] (JSD) between generated shapes by ADV3D and vertex deformation (VD). Specifically, we calculate JSD by uniformly sampling point clouds of 1000 points from the optimized shapes with the birds-eye-view 2D histogram of all CAD models in our asset sets (resolution of $100 \times 100$).

## A.6 Additional Discussions

**Offroad or Collision Evaluation for Planning?** We do not adopt offroad and collision as adversarial objectives or evaluation metrics. This is because the planner [24] leverages map information to prevent offroad trajectories to ensure safety driving. Moreover, adding a collision loss requires careful formulation to provide stable supervision (such as through actor-ego distance or time-to-collision), as the vanilla implementation results in a sparse and noisy objective function to optimize. To ease optimization stability and enable fast convergence, we designed our objective function to leverage losses that can be easily computed for perception, prediction, and planning at each timestep over the full scenario. We also do not modify actor behavior, making it challenging to generate collision outcomes in the SDV on real-world highway driving scenarios.

**Gradient-based attacks for Adv3D?** Gradient-based attacks require fully differentiable autonomy and simulation systems. Since the goal of Adv3D is to cater to any autonomy system, not just those that are fully differentiable, we chose the black-box algorithm BO to ensure broader applicability. For systems that are completely differentiable, one may pursue gradient-based white-box attacks with a differentiable simulator, which can be stronger and more efficient.

# B Additional Results and Analysis

**Attacking Full Autonomy Stack:** To take into account the full autonomy stack, we find it is important to use an adversarial objective that takes a combination of submodule costs. In Tab. A7, we provide the full results for Tab. 3 in the main paper, including missing combinations $\mathcal{M}_4$ and $\mathcal{M}_5$. Moreover, we also compare with the results in closed loop using shapes generated by open-loop attack.

**Latent Asset Representation:** We repeat the experiment from Tab. 3 but now using a lower density 100 triangle mesh which has a lower dimension thus can be optimized with BO. Results in Tab. A8 show that large vertex deformation is required to achieve similar attack strength as Adv3D. Moreover, the vertex deformed actors are overly simplified and unrealistic with noticeable artifacts.

---

[3]https://github.com/microsoft/FLAML

| #ID | Opt. Settings | Perception $\sum_t \ell^t_{det}$ | Prediction $\sum_t \ell^t_{pred}$ | Planning $\sum_t c^t_{plan}$ | AP↑ (%, @0.5) | Recall↑ (%, @0.5) | minADE↓ $L_2$ error | meanADE↓ $L_2$ error | Lat.↓ $(m/s^2)$ | Jerk↓ $(m/s^3)$ |
|---|---|---|---|---|---|---|---|---|---|---|
| Original | | | | | 88.7 | 89.4 | 2.51 | 4.99 | 0.261 | 0.294 |
| $\mathcal{M}_1$ | Open-Loop | ✓ | | | 80.4 | 87.6 | 2.01 | 4.95 | 0.256 | 0.301 |
| | Closed-Loop | | | | **69.6** | **71.4** | 1.97 | 5.02 | 0.239 | 0.310 |
| $\mathcal{M}_2$ | Open-Loop | | ✓ | | 83.5 | 88.5 | 2.52 | 5.39 | 0.223 | 0.341 |
| | Closed-Loop | | | | 83.1 | 89.1 | 2.92 | **6.34** | 0.254 | 0.412 |
| $\mathcal{M}_3$ | Open-Loop | | | ✓ | 87.2 | 88.8 | 2.57 | 5.38 | 0.305 | 0.352 |
| | Closed-Loop | | | | 86.7 | 88.3 | 2.94 | 6.03 | 0.324 | **0.434** |
| $\mathcal{M}_4$ | Open-Loop | ✓ | ✓ | | 79.9 | 85.9 | 2.57 | 5.35 | 0.231 | 0.353 |
| | Closed-Loop | | | | 70.1 | 78.8 | 2.90 | 5.98 | 0.223 | 0.401 |
| $\mathcal{M}_5$ | Open-Loop | ✓ | | ✓ | 81.2 | 84.3 | 2.57 | 5.60 | 0.333 | 0.253 |
| | Closed-Loop | | | | 72.3 | 75.0 | **2.95** | 6.04 | 0.342 | 0.401 |
| $\mathcal{M}_0$ | Open-Loop | ✓ | ✓ | ✓ | 85.5 | 87.7 | 2.73 | 5.99 | 0.262 | 0.372 |
| | Closed-Loop | | | | 75.4 | 76.4 | 2.82 | 6.21 | **0.411** | 0.410 |

Table A7: **Full table of adversarial optimization for the full autonomy stack.** Unlike existing works that consider sub-modules, ADV3D generates actor shapes that are challenging to all down-stream modules.

| Algorithms | AP ↑ (%, @0.5) | Recall ↑ (%, @0.5) | minADE ↓ $L_2$ error | Jerk ↓ $(m/s^3)$ | JSD |
|---|---|---|---|---|---|
| Original | 98.7 | 99.6 | 4.70 | 0.090 | – |
| VD: $0.05m$ | 98.7 | 99.6 | 5.04 | 0.090 | 0.057 |
| VD: $0.1m$ | 98.7 | 99.6 | 5.10 | 0.090 | 0.137 |
| VD: $0.2m$ | 98.7 | 99.6 | 5.16 | 0.090 | 0.253 |
| VD: $0.5m$ | 78.3 | 81.2 | 5.77 | 0.103 | 0.745 |
| VD: $1.0m$ | 45.5 | 48.5 | 5.79 | 0.155 | 0.758 |
| ADV3D (ours) | 50.3 | 55.8 | 7.87 | 0.111 | 0.175 |

Table A8: Compare with vertex deformation.

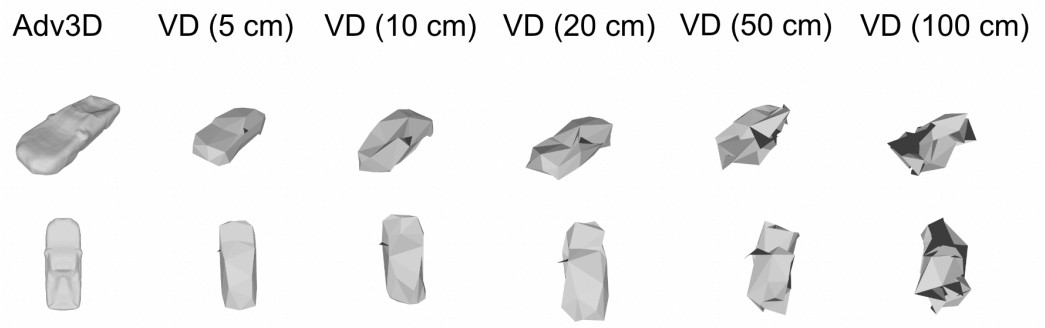

Figure A6: Qualitative comparisons with Vertex-Deformation (VD). Top and bottom show side and top-down views respectively.

**Additional Qualitative Examples:** We provide more qualitative examples in Figure A7, A8 and A9 to show that ADV3D is able to generate safety-critical actor shapes for autonomy testing with appearance coverage.

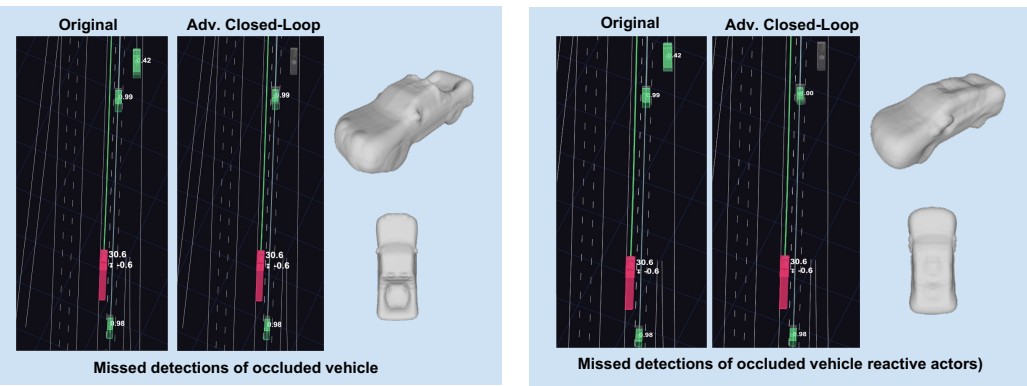

Figure A7: Qualitative examples of adversarial shape generation (non-reactive actors vs reactive actors). ADV3D is able to generate adversarial actors that cause detection failures due to occlusion.

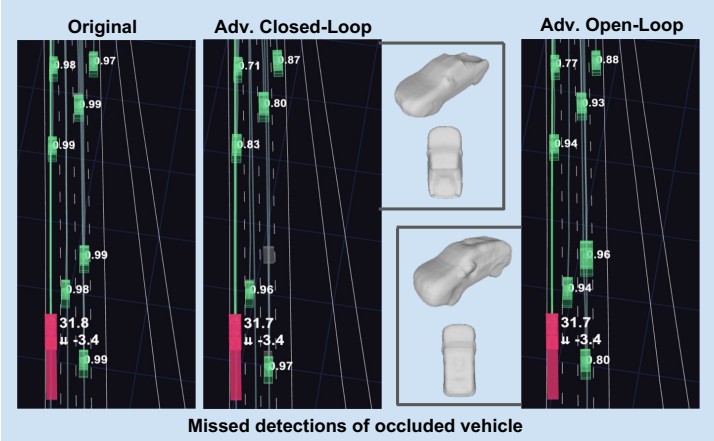

Figure A8: Qualitative examples of adversarial shape generation in closed loop vs open loop. ADV3D is able to generate adversarial actors that cause detection failures due to occlusion but the open-loop counterpart fails.

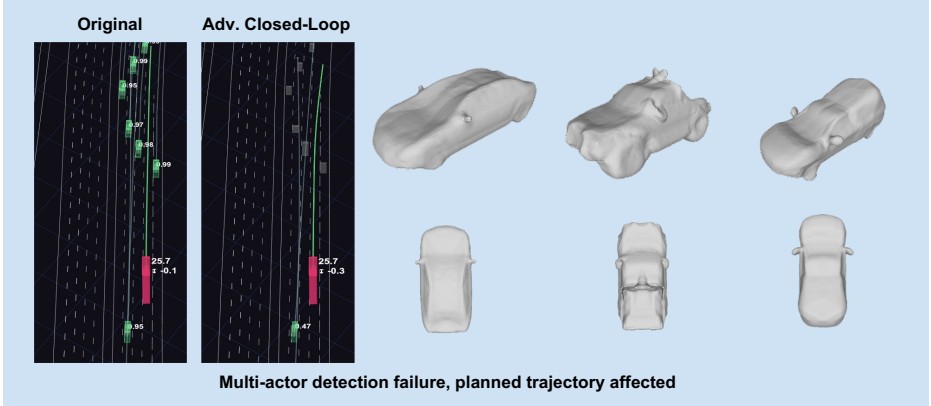

Figure A9: Qualitative examples of adversarial shape generation with multi-actor attacks. ADV3D creates three safety-critical shapes that cause detection failures for all front actors.

