# OpenReview forum: "Adv3D: Generating Safety-Critical 3D Objects through Closed-Loop Simulation"
_robot-learning.org/CoRL/2023/Conference — CoRL 2023 Poster_

### Official Review · Reviewer_nxxv · 2023-07-11

**Confidence:** 1
**Originality:** Good
**Technical Quality:** Good
**Clarity Of Presentation:** Fair
**Impact:** 2

**Recommendation:**

Weak Accept: I recommend accepting the paper, but will not argue for my recommendation if the majority of other reviewers have a different opinion.

**Review:**

The problem studied in this paper is interesting. The supplementary materials and the video are helpful to understand the proposed algorithm.

Some comments are as follows:
- In Sec. 3.2, the actor is placed using the pose based on S_t. Does this pose is known or it needs to be optimized?
- Table 2 and Table 3 are a bit confusing. It seems that the proposed method does not outperform existing methods in general.
- Typo in Sec. 3.2 paragraph 3.

**Quality Of The Limitations Section:**

Additional details required

**Questions For Rebuttal:**

Please refer to my review comments above.

**Robotics Focus:**

Highly relevant to robotics but no hardware experiments

**Summary Of Paper:**

This paper proposes a framework (ADV3D) that performs closed-loop sensor simulations for evaluating self-driving vehicles (SDVs). This framework focuses on LiDAR-based data. The proposed framework utilizes a low-dimensional shape representation to modify the vehicle shape. The experiments show that ADV3D is able to generate challenging actor shapes. Moreover, it is shown that the closed-loop is more powerful than attacking in the open-loop.

**Summary Of Recommendation:**

Please refer to my review comments above.

---

### Official Review · Reviewer_xR6w · 2023-07-15

**Confidence:** 4
**Originality:** Good
**Technical Quality:** Good
**Clarity Of Presentation:** Very Good
**Impact:** 3

**Recommendation:**

Weak Reject: I recommend rejecting the paper, but will not argue for my recommendation if the majority of other reviewers have a different opinion.

**Review:**

I summarize the strengths and weaknesses of the paper below.

**Strength**:

1. The paper investigates an important problem (i.e. adversarial attacking) in self-driving to evaluate the robustness of autonomous driving systems.

2. The idea of evaluating the performance of the system in a closed-loop manner is insightful.

3. The experiments are well designed to demonstrate the effectiveness of various components of the method.

**Weaknesses** and **Questions**:

1. The main contribution of the paper seems to be incremental: a) a simple PCA-based latent space. b) a loss function taking into account the future trajectory of SDV induced by the modified vehicle shape.

2. The definition of $W$ in Eq (2) is unclear. I suppose it is the PCA components computed from the whole CAD dataset?

3. The selection of the PCA-based latent space of the vehicle shape needs more justification. The encoder/decoder in Eq (2) seems to not guarantee obtaining the same/similar output shape from a given input shape since $W$ is not invertible. Furthermore, how does this simple encoder/decoder approach compare to more sophisticated neural network based models (VAE)? It is necessary to include such experiments to validate the selection of the PCA-based latent space.

4. The reason for selecting the closest actor in front of the SDV to be modified needs more discussion. The agents aside the SDV might also have a significant impact in the system's performance. Optimizing the selection of the modified actor(s) can be a valuable option.

5. Figure 4 is hard to interpret. It is unclear how the green and blue arrows related to perception and prediction failures. And the numbers in the figure are also not defined.

6. The comparison against vertex-wise noise baseline would be more meaningful if the average performance over the whole dataset is reported. Currently, it is biased since it only compares the methods on the ADV3D's best data. Also, did the authors also apply optimization over the vertex-wise noise for the baseline?

7. The advantage of the proposed approach over the existing method is unclear. From Table 2, while the proposed method is averagely good across metrics, it is not the best one in most of them. Can we achieve better results by simply combining existing methods?

8. How is the scalability of the method to multiple modified actors? In other words, hoe many optimization iterations are required to achieve a good result (compared to the baseline) when the number of modified actors increases?



**Quality Of The Limitations Section:**

Limitations are addressed clearly

**Questions For Rebuttal:**

See the questions raised in the above section.

**Robotics Focus:**

Highly relevant to robotics but no hardware experiments

**Summary Of Paper:**

This paper introduces a novel framework for adversarially attacking self-driving vehicle autonomy systems, ADV3D. It focuses on LiDAR-based systems and tried to attack and evaluate the full stack of self-driving autonomy systems, including perception, prediction, and planning. It attacks the system by generating realistic adversarial vehicle shapes around the autonomous agent. The adversarial vehicle shape is optimized in the latent space through a simple PCA-based vehicle shape encoder-decoder framework. Unlike previous methods, the proposed optimization takes into account the so-called closed-loop performance loss of the autonomous vehicle simulation. The experiments demonstrate its effectiveness in finding worse failure cases than baseline approaches in a captured dataset including 100 real autonomous driving scenes.

**Summary Of Recommendation:**

I am slightly on the negative side of the recommendation due to the weaknesses and questions I have raised above. The major concerns are the significance of the contribution and the justification of the algorithm design choice.

---

### Official Review · Reviewer_BYfr · 2023-07-20

**Confidence:** 3
**Originality:** Very Good
**Technical Quality:** Excellent
**Clarity Of Presentation:** Very Good
**Impact:** 3

**Recommendation:**

Strong Accept: I recommend accepting the paper and will argue for my recommendation even if other reviewers hold a different opinion.

**Review:**

Overall I think the paper is solid and well-written. The approach makes sense and the engineering effort is impressive. While closed-loop simulation remains challenging for a gradient-based approach to train on, the BO approach can certainly work. The SDF-then-PCA approach for generating adversarial shape is interesting and novel albeit simple. The experiments are well-designed with a high level of engineering implementation. The result shows the benefit of the approach, although I do have some questions (see next section).

**Quality Of The Limitations Section:**

Limitations are addressed clearly

**Questions For Rebuttal:**

1. There are two autonomy stacks mentioned, yet in the result section they are not discussed separately, what is the actual setup?
2. In table 1, it seems that open-loop attack sometimes result in better score than the baseline, can you explain why?
3. When evaluating planning, my intuition is to use offroad or collision as the primary metrics, can the authors explain the reason behind their choice?
4. BO can be applied to any black-box system, yet it is not scalable, can the authors comment on the practical use of BO v.s. gradient-based adversarial attack approaches?


**Robotics Focus:**

Highly relevant to robotics but no hardware experiments

**Summary Of Paper:**

This paper presents an adversarial shape attack pipeline that targets the whole autonomous vehicle stack consisting of perception, prediction, and planning. The adversarial shape is searched by mapping a suite of common shapes into a latent space and performing a search there. Specifically, the authors starts from  CAD assets and turn them into SDFs, then apply PCA on the SDFs to obtain a latent representation together with the principal axes.

The simulation environment uses raycasting to simulate Lidar point cloud, then the AV either takes an instance-based stack to predict the environment motion and plan its own motion, or takes an implicit occupancy-based approach to plan its motion. The other agents follow a traffic model.

The adversarial search uses a Bayesian optimization approach that targets the whole stack and the result shows that the proposed approach is able to generate more realistic adversarial shapes than the baseline, and closed-loop training is superior.

**Summary Of Recommendation:**

I think this paper proposes an interesting approach to adversarial attacks on AV. Particularly, the PCA approach that uses actual CAD assets is novel and significantly improves the realism of the adversarial shape attack. The engineering effort is commendable given that the authors built a functional closed-loop simulator with sensor simulation, a complete AV stack, and a traffic model for other agents. The results are nicely explained and make sense to me. Overall, I find this paper well-written, the research of high quality, and suitable to be published in CoRL.

---

### Official Review · Reviewer_htaN · 2023-07-22

**Confidence:** 3
**Originality:** Good
**Technical Quality:** Fair
**Clarity Of Presentation:** Good
**Impact:** 3

**Recommendation:**

Weak Reject: I recommend rejecting the paper, but will not argue for my recommendation if the majority of other reviewers have a different opinion.

**Review:**

Strenghts
=================
- The use of a low-dimensional latent code for object shape parameterization and restricting the search space to realistic vehicle shapes is an elegant approach to mining difficult examples for stress-testing autonomous vehicles.
- In general, the paper is written clearly and reads well.
- The figures are well made and illustrate the generated adversarial shapes well.

Constructive Criticism
====================
- The paper states that their method attacks the full autonomy stack. This statement is misleading in my opinion. The presented approach attacks the perception stack and closes the loop in simulation. Just closing the loop does not mean it attacks the full autonomy stack. Attacking the full autonomy stack would also include alterations to simulation such as the behavior of other road users or the dynamics of the simulated vehicle.

- In Table 2, what method is being tested in this table? From my understanding, each row corresponds to a method that manipulates the LIDAR data, but what approach is used for perception and planning? Why do some rows contain contradicting results? For example: the second row (M1) results in very low AP and Recall, but performs best in lateral acceleration and second-to-best in jerk. It would be interesting to understand where this discrepancy is coming from. Extending the caption of table 2 to be more self-contained would help the reader understand the presented data. In general, more interpretation of the presented results would greatly improve the paper IMO.

- The adversarial shape modification is based on a dataset of real-world LIDAR data and CAD assets. It would be interesting to understand how the performance obtained from the generated adversarial shapes compares to the performance obtained when using the original assets. Do the adversarial shapes degrade performance more than any shape existing in the dataset?

- This is a very minor comment, but I wouldn't call 2000s or highway data a large-scale dataset.

**Quality Of The Limitations Section:**

Limitations are addressed clearly

**Questions For Rebuttal:**

- The adversarial attacks proposed in this paper are focused on the perception system of a LIDAR-based autonomous vehicle. While the paper mentions the importance of closed-loop evaluation, it is not fully clear to me why this is the case. Why can the perception system not be evaluated on a dataset without closing the loop? Where does the difference between open-loop evaluation and closed-loop evaluation come from? One hypothesis might be that closing the loop results in novel viewing angles that result in poor performance when combined with adversarial shapes. It would be interesting to better understand why closed-loop perception is important in this application.

- SOTA autonomous driving systems combine multiple sensor modalities for perception, such as cameras, radar, and LIDAR. One of the challenges that these systems face are for example challenging edge cases (for example: driving on a highway behind a car that is carrying a bike). Could the proposed approach be extended to generate LIDAR readings of such challenging scenarios?

**Robotics Focus:**

Relevant but unlikely to deploy to hardware in near future

**Summary Of Paper:**

This paper proposes Adv3D, a framework to stress-test autonomous driving approaches by generating challenging shapes of the road user immediately in front of the autonomous vehicle in a simulator. In doing so, Adv3D challenges LIDAR-based perception stacks as it manipulates the appearance towards shapes that harm the performance of the autonomy stack.

The shape manipulation is performed using adversarial shape optimization, where a learned latent 3D encoding of a vehicle is iteratively updated using Bayesian optimization such that the overall autonomy performance drops significantly in simulation.

The proposed approach is evaluated on a dataset containing 100 driving sequences, each 20s in duration. Each segment is replayed in a simulator while the LIDAR data is modified. To measure the impact of this altered sensor data, two perception systems are tested, both showing degraded performance when deployed on manipulated data.

**Summary Of Recommendation:**

The basis for my recommendation is comprised of the following observations:
- The paper provides a limited interpretation of the presented results. Especially if the results are contradicting (such as Table 2), the paper should provide more interpretation and analysis of the obtained results.
- The paper makes the misleading claim of attacking the entire autonomy stack, even though the presented adversarial attacks are limited to perception.

---

### Author Response · Authors · 2023-08-10
**General Response**

We thank the reviewers for their thoughtful reviews and valuable comments. We are excited that the reviewers believed we investigate an “interesting” and “important” problem [**Reviewer BYfr, Reviewer xR6w, Reviewer nxxv**], appreciated that our idea is “insightful” and “novel” [**Reviewer BYfr, Reviewer xR6w**], and acknowledged our experiments / results are “well designed / explained” [**Reviewer BYfr, Reviewer xR6w**]. We summarize the major points as follows.

**Clarification of Table 2 [Reviewer htaN, Reviewer xR6w, Reviewer nxxv]**  \
In Table 2, each row corresponds to a different adversarial objective we used for Adv3D. Specifically, $\mathcal{M}_1, \mathcal{M}_2, \mathcal{M}_3$ represents attacking perception, prediction and planning modules separately. “Original” corresponds to the original scenario without modified actor shape. As shown in Table 2, attacking each downstream module produces challenging objects that are only risky to that module. For instance, $\mathcal{M}_1$ degrades perception performance but does not harm prediction and planning performance.  This demonstrates that attacking the full autonomy system is critical. As suggested by Reviewer htaN, we have revised Table 2 to be more self-contained to avoid possible confusion and misunderstandings. The “Prior works” column was meant to refer to past works that only attacked perception, prediction, or planning (each row has a similar loss formulation) - we have removed this column.

**Novelty and technical contribution [Reviewer xR6w]** \
Our paper’s novelty lies in developing a system to generate safety-critical actor shapes for autonomy testing through closed-loop simulation of real-world traffic scenarios. We believe Adv3D is a critical and innovative step towards more realistic and practical adversarial robustness testing for robotics. Exploiting existing algorithms  to realize a novel idea does not mean there is no technical contribution (as recognized by Reviewer BYfr). Please refer to the individual response for more details.

We now address the concerns of each reviewer individually. We have also included a **revised pdf (changes highlighted in blue)**. We look forward to follow up discussions!

---

### Author Response · Authors · 2023-08-15
**Looking forward to follow-up discussion**

Dear ACs,

Thank you so much for the efficient handling of our submission and leading the rebuttal discussion. Since the deadline of discussion is approaching, we look forward to any follow-up discussion needed by reviewers.

So far, we received two positive scores (strong accept, weak accept) from Reviewer BYfr and Reviewer nxxv and 2 negative scores (weak reject, weak reject) from Reviewers htaH and Reviewer xR6w. The main critique lies at some possible misunderstandings on our contributions and experiment setups/results. To alleviate reviewers' concerns, we made a significant effort to clarify our novelty [Reviewers xR6w], experiment results in Table 2 [Reviewer htaN, Reviewer xR6w, Reviewer nxxv] and conduct new experiments suggested by [Reviewers htaN, Reviewer xR6w].

We sincerely hope that all reviewers could take a close look at our response and find it convincing. Please do not hesitate to contact us if reviewers have further questions and comments.

Thank you for your time and help!

Best, \
Authors

---

### Decision · Program_Chairs · 2023-08-30

**Decision:**

Accept (Poster)

**Comment:**

The reviews for this paper were a bit diverging. The reviewers generally agree that this paper addresses a timely problem (namely, that of stress-testing AV stacks) and is well executed, but also raise a number of concerns, in particular in terms of technical novelty. Upon reading the paper, the AC believes that the "system-level" contribution of this paper is indeed notable and outweighs its rather limited technical contribution. Thus, the AC recommends acceptance. In the final version of this paper, the authors should carefully revise the paper to address all reviewers' comments / suggestions.